# Statistical Optimisation of Phenol Degradation and Pathway Identification through Whole Genome Sequencing of the Cold-Adapted Antarctic Bacterium, *Rhodococcus* sp. Strain AQ5-07

**DOI:** 10.3390/ijms21249363

**Published:** 2020-12-09

**Authors:** Gillian Li Yin Lee, Nur Nadhirah Zakaria, Peter Convey, Hiroyuki Futamata, Azham Zulkharnain, Kenshi Suzuki, Khalilah Abdul Khalil, Noor Azmi Shaharuddin, Siti Aisyah Alias, Gerardo González-Rocha, Siti Aqlima Ahmad

**Affiliations:** 1Department of Biochemistry, Faculty of Biotechnology and Biomolecular Sciences, Universiti Putra Malaysia, Serdang UPM 43400, Selangor, Malaysia; gillian5332@gmail.com (G.L.Y.L.); nadhirahairakaz@gmail.com (N.N.Z.); noorazmi@upm.edu.my (N.A.S.); 2British Antarctic Survey, NERC, High Cross, Madingley Road, Cambridge CB3 0ET, UK; pcon@bas.ac.uk; 3Graduate School of Science and Technology, Shizuoka University, Hamamatsu 432-8561, Japan; futamata.hiroyuki@shizuoka.ac.jp; 4Research Institute of Green Science and Technology, Shizuoka University, Suruga-ku, Shizuoka 422-8529, Japan; suzuki.kenshi.15@shizuoka.ac.jp; 5Department of Bioscience and Engineering, College of Systems Engineering and Science, Shibaura Institute of Technology, 307 Fukasaku, Minuma-ku, Saitama 337-8570, Japan; azham@shibaura-it.ac.jp; 6School of Biology, Faculty of Applied Sciences, Universiti Teknologi MARA, Shah Alam 40450, Selangor, Malaysia; khali552@uitm.edu.my; 7National Antarctic Research Centre, B303 Level 3, Block B, IPS Building, Universiti Malaya, Kuala Lumpur 50603, Malaysia; saa@um.edu.my; 8Institute of Ocean and Earth Sciences, B303 Level 3, Block B, Universiti Malaya, Lembah Pantai, Kuala Lumpur 50603, Malaysia; 9Laboratorio de Investigacion en Agentes Antibacterianos, Facultad de Ciencias Biologicas, Universidad de Concepcion, Concepcion 4070386, Chile; ggonzal@udec.cl

**Keywords:** Antarctica, *Rhodococcus*, next-generation sequencing, de novo assembly, β-keto-adipate pathway

## Abstract

Study of the potential of Antarctic microorganisms for use in bioremediation is of increasing interest due to their adaptations to harsh environmental conditions and their metabolic potential in removing a wide variety of organic pollutants at low temperature. In this study, the psychrotolerant bacterium *Rhodococcus* sp. strain AQ5-07, originally isolated from soil from King George Island (South Shetland Islands, maritime Antarctic), was found to be capable of utilizing phenol as sole carbon and energy source. The bacterium achieved 92.91% degradation of 0.5 g/L phenol under conditions predicted by response surface methodology (RSM) within 84 h at 14.8 °C, pH 7.05, and 0.41 g/L ammonium sulphate. The assembled draft genome sequence (6.75 Mbp) of strain AQ5-07 was obtained through whole genome sequencing (WGS) using the Illumina Hiseq platform. The genome analysis identified a complete gene cluster containing *catA*, *catB*, *catC*, *catR*, *pheR*, *pheA2,* and *pheA1*. The genome harbours the complete enzyme systems required for phenol and catechol degradation while suggesting phenol degradation occurs via the β-ketoadipate pathway. Enzymatic assay using cell-free crude extract revealed catechol 1,2-dioxygenase activity while no catechol 2,3-dioxygenase activity was detected, supporting this suggestion. The genomic sequence data provide information on gene candidates responsible for phenol and catechol degradation by indigenous Antarctic bacteria and contribute to knowledge of microbial aromatic metabolism and genetic biodiversity in Antarctica.

## 1. Introduction

The last largely pristine and remote wilderness on Earth, Antarctica is facing a variety of anthropogenic impacts associated with rapidly increasing human activities. The first arrival of explorers and extraction of marine living resources in Antarctica and the Southern Ocean region date back to the 18th Century [1,2]. Today, while the focus of human presence has shifted from imperialism and exploitation to scientific research and tourism, the number of visitors continues to increase, leading to multiple visible human impacts on the Antarctic environment [1,3,4]. Of all “local” impacts within Antarctica, oil spills have been considered amongst the most damaging through the persistence of petroleum products in this cold environment [5,6]. Recent studies have reported instances of pollution by hydrocarbons and associated compounds, including polycyclic aromatic hydrocarbons, chlorinated biphenyls, and phenols [7,8,9]. Due to the chronically low temperatures and dry or frozen conditions across much of Antarctica, the continent’s ecosystems are very sensitive to even small environmental changes, and aromatic hydrocarbon products such as phenol and phenolic compounds (PCs) can have considerable detrimental impacts on terrestrial and aquatic life [10,11,12].

Phenol is a component present in diesel, petrol, and lubricant oils, as well as being used in the preparation of pesticides, herbicides, bactericides, and fungicides [13]. Due to its anti-microbial properties, phenol is also often used as an antiseptic in medical and cosmetic industries. Phenol poses serious environmental concerns due to its toxicity towards living organisms (including humans) even at low concentration [14,15].

The process of bioremediation utilises microorganisms to transform toxic compounds into less or non-toxic toxic forms. Bioremediation is generally more efficient in degrading organic compounds, often completely, in comparison with physicochemical treatments [16,17]. Members of the bacterial genera *Pseudomonas*, *Arthrobacter*, *Rhodococcus,* and *Acinetobacter*, capable of degrading phenol, have been reported [18,19,20,21].

About 90% of the Earth’s ocean volume has a temperature of 5 °C or less, and over 80% of the Earth’s surface is permanently cold when the terrestrial habitats of the polar regions and areas such as the Tibetan plateau are included [22,23]. Numerous phenol-contaminated sites are characterised by low temperature [24,25]. Biodegradation of phenol in cold regions requires cold-adapted or cold-tolerant microorganisms, but the metabolic activities of most mesophilic phenol-degrading microorganisms studied to date are severely limited by low temperature [19,26,27]. Antarctic microorganisms are known to be adapted to the harsh conditions of Antarctica, including low temperature, high solar radiation, and low nutrient availability [10]. The identification of indigenous microorganisms capable of degrading pollutants, including hydrocarbons, could be particularly important for future environmental management and bioremediation procedures in Antarctica, particularly because the very strict environmental regulations applied under the Antarctic Treaty are likely to prevent the introduction of non-native species for this purpose [2,28,29].

*Rhodococcus* sp. strain AQ5-07, originally isolated from non-human-impacted soil obtained on King George Island (South Shetland Islands), is capable of degrading 0.5 g/L of phenol within 96 h at 10 °C [30]. In this study, we report that the optimum temperature predicted by response surface methodology (RSM) was in the range of optimum temperature revealed by one-factor-at-a-time (OFAT) analysis. The highest phenol degradation of 94.11% was achieved under conditions predicted by RSM within 84 h at 12.5 °C. We additionally report the draft genome of strain AQ5-07 and identify gene candidates responsible for phenol degradation. We also verified ring cleavage activities in the catechol degradation pathway to support the involvement of gene candidates in complete phenol metabolism.

## 2. Results

### 2.1. Plackett-Burman Design

The Plackett–Burman design applied to strain AQ5-07 (Table 1) generated phenol degradation levels from 44.12% to 61.08% in the 12 runs. Maximum phenol degradation was achieved at 25 °C, pH 6, and NaCl and ammonium sulphate concentrations of 0.3 and 0.5 g/L, respectively. The lowest phenol degradation was observed at 5 °C, pH 8, and NaCl and ammonium sulphate concentrations at 0.30 and 0.1 g/L, respectively.

ANOVA (Table 2) confirmed that the model was significant (*p* = 0.034) and highly reliable. Temperature (A), pH (C), and ammonium sulphate concentration (D) were the significant factors affecting phenol degradation by strain AQ5-07, while NaCl concentration (and all interaction terms) had no significant (*p* > 0.05) influence and was excluded in the subsequent central composition design (CCD) experiment.

### 2.2. CCD

CCD was employed to study the interactions between the three significant factors and to determine the optimum conditions for phenol degradation by strain AQ5-07. Table 3 shows the experimental matrix of CCD with the corresponding experimental and predicted values of phenol degradation by strain AQ5-07. The highest experimental and predicted values of phenol degradation were 94.10% and 93.45%, respectively, at 12.5 °C, pH 7, and 0.4 g/L ammonium sulphate. The lowest phenol degradation values (experimental, 2.76%, predicted 3.43%) were obtained at 0 °C, pH 7, and 0.4 g/L ammonium sulphate. ANOVA (Table 4) revealed that the model was highly significant (*p* < 0.0001).

The 3D response surface was plotted using Design-Expert software version 6 to visualise the interaction effects of the significant variables. Each figure represents the interaction effects between two independent variables while holding the other variable at constant level. Figure 1 illustrates the interaction of temperature and pH, Figure 2 that between temperature and ammonium sulphate concentration, and Figure 3 the interaction between pH and ammonium sulphate concentration.

Figure 1 shows that the optimum conditions for phenol degradation by strain AQ5-07 were a temperature between 12.5 and 16.25 °C and pH between 6.5 and 7.5. Based on Figure 2, optimum phenol degradation was again achieved between 12.5 and 16.25 °C and ammonium sulphate concentration between 0.35 and 0.45 g/L. Figure 3 indicates optimum degradation between pH 6.5 and pH 7.5 and ammonium sulphate concentration between 0.35 and 0.45 g/L.

### 2.3. Validation Experiment

The three significant variables were maintained at optimum values of 14.8 °C, pH 7.05, and 0.41 g/L ammonium sulphate in order to experimentally test the predicted value of phenol degradation. The experimental value obtained was 92.91% phenol degradation, close to the model’s predicted value (94.61%), supporting the validity of the model.

### 2.4. Genomic Features of Rhodococcus sp. Strain AQ5-07

The genome of strain AQ5-07 was sequenced using the Illumina HiSeq 2500-PE125 platform. The draft genome obtained was 6,749,221 bp long with GC content of 62.4%. Key genomic features of strain AQ5-07 are summarised in Table 5. The assembled genome was arranged in 34 contigs, with the length of the longest contig being 1,378,314 bp. A total of 6545 coding sequences (CDSs), four rRNAs, and 75 tRNAs were predicted using the Rapid Annotations using Subsystems Technology (RAST) server with the SEED database, which contains accurate and up-to-date annotations for microbial genomes. The Whole Genome Shotgun project was annotated using NCBI Prokaryotic Genome Annotation Pipeline.

Based on RAST annotation of the draft complete genome, coding sequences were grouped into 415 subsystems. The number of genes in each subsystem and the subsystem coverage are shown in Figure 4, with overall 35% of the total CDSs classified into subsystems and 65% of CDSs excluded. The subsystem of amino acids and derivatives contained the highest number of coding sequence (686 counts), followed by 574 counts in the subsystem of carbohydrates and 438 counts in the subsystem of cofactors, vitamins, prosthetic groups, and pigments. The genome also included one antifreeze protein, six cold shock proteins, and 88 genes associated with the metabolism of aromatic compounds.

### 2.5. Identification of Gene Candidates for Phenol Degradation

Bioinformatics analyses were carried out to identify the gene candidates responsible for phenol degradation. The gene-associated functions of phenol and catechol catabolic genes were predicted using the NCBI genomic database and RAST server with SEED, Kyoto Encyclopedia of Genes and Genomes (KEGG), and Clusters of Orthologous Groups (COG) databases (Table 6). Figure 5 illustrates the physical map of the gene cluster containing phenol hydroxylase and other enzymes including catechol 1,2-dioxygenase (C12D) involved in the catechol degradation pathway. The gene cluster from strain AQ5-07 is almost identical to that of the well-studied *Rhodococcus erythropolis* CCM2595. When compared to the gene cluster from *Rhodococcus jostii* RHA1, generally the gene structure is highly conserved in strain AQ5-07, with the exception of the 2.9 kbp insertion between the *cat* gene and the *phe* gene cluster. The CatRABC operon was identified within the gene cluster, which is involved in the *ortho*-pathway of catechol metabolism. Genes encoding catechol 2,3-dioxygenase (C23D) and downstream enzymes related to *meta*-pathway were not detected in the genome. Contiguous to the *cat* operon, the *phe*R-*phe*A2A1 gene cluster was predicted, shown in Figure 5.

### 2.6. Enzymatic Assay for catechol 1,2-dioxygenase (C12D) and catechol 2,3-dioxygenase (C23D)

Bioinformatics analyses predicted the existence of a single copy of the *catA* gene, encoding C12D in strain AQ5-07. To determine whether other catechol ring cleavage enzymes are available, enzymatic assays for C12D and C23D were performed using cell-free extracts. Figure 6 illustrates the outcome of assays of the production of cis, cis–muconic acid (CCMA) by C12D and 2-hydroxymuconic semialdehyde (2-HMS) by C23D, respectively. The concentration of CCMA slowly increased throughout the assay, reaching 46.55 μM. However, no production of 2-HSM was detected, supporting only the presence of C12D activity in strain AQ5-07. The average rate of CCMA production over the 1 h assay was 0.78 µM/min. The calculated maximum specific activity of C12D was 23.603 U/mg after 10 min of incubation.

## 3. Discussion

In general, the OFAT approach is less sensitive in optimisation, as it does not include interaction effects between variables, potentially leading to misinterpretation of the results obtained [30]. Statistical optimisation by RSM optimises all the significant parameters effectively [31]. RSM is commonly used to evaluate the relationships between several explanatory variables and the response variables [32]. Previously, Lee et al. [30] reported that *Rhodococcus* sp. strain AQ5-07 achieved a highest phenol degradation of 89.93% under OFAT at 10.0 °C, pH 7.0, and 0.3 g/L NaCl. Therefore, in this study, a Plackett–Burman design and CCD were employed to provide an effective means of optimizing the conditions for phenol degradation by strain AQ5-07 under the three significant independent variables of temperature, pH, and ammonium sulphate concentration. The elliptical shapes of the 3D response surfaces shown Figure 1 and Figure 2 and the circular shape of the response surface in Figure 3 indicate interaction effects between the two variables in each figure, with the peaks suggesting the mutual relationships between these variables and the centre of the system that represents the point where maximum phenol degradation was achieved [33,34]. The optimum temperature for phenol degradation, in the range of 12.5 to 16.25 °C, is consistent with the strain being psychrotolerant [35,36]. The results obtained were consistent with the current study’s data. The optimum temperature predicted by RSM was in the range of optimum temperature revealed by OFAT. However, optimum concentrations of NaCl and pH as predicted by RSM were higher than those of OFAT. The highest phenol degradation of 94.11% was achieved under conditions predicted by RSM of 12.5 °C, pH 7.0, and 0.4 g/L NaCl.

Next generation sequencing and application of bioinformatics give insights into the metabolic pathways of microorganisms [37,38]. Progressive advances in next generation sequencing technologies have led, for instance, to the characterisation of novel biochemical pathways of biogeochemical significance and insights into the phylogenetic and the functional diversity of hydrocarbon-degrading microorganisms [39]. According to Ehsani et al. [40], members of the bacterial genus *Rhodococcus* are Gram-positive and high GC actinomycetes. The genomes of several *Rhodococcus* species with >60% GC content have been reported in recent studies, as found here in strain AQ5-07 [40,41,42]. Similar results have been reported in a very recent sub-Antarctic study [43] where the genomes of two *Rhodococcus* species both contained 62.3% GC.

Genome annotation showed that the most abundant genes classified into subsystems were related to metabolism of amino acids and derivatives, carbohydrates, cofactors, vitamins, prosthetic groups, and pigments, all of which are essential for cell survival [44]. Temperature is one of the major environmental challenges facing microbial life in Antarctica [45]. In the genome, 151 stress response proteins were annotated including proteins with roles in responding to osmotic stress as well as cold and heat shock. The genome of strain AQ5-07 contains a total of six genes encoding for cold shock proteins with four of them categorised as cold shock protein A and one as cold shock protein C. Most bacteria produce cold shock proteins to counteract the harmful effects of declining temperature and to enable growth at low temperatures [46,47]. A gene coding for a type I antifreeze protein was also detected based on RAST annotation, showing 75.5% amino acid sequence identity with type I antifreeze protein from *Rhodococcus* spp. (accession no. WP_006944967). Part of the survival strategy employed by microorganisms in Antarctic habitats includes the production of antifreeze proteins [48]. Similarly, Gilbert et al. [49] reported that a hyperactive, Ca^2+^-dependent, antifreeze protein in the Antarctic bacterium *Marinomonas primoryensis* might be capable of preventing lethal freezing in ice-covered Antarctic lakes.

Members of the genus *Rhodococcus* are well known for their metabolic capabilities for biodegradation of various environmental pollutants, including aliphatic and aromatic hydrocarbons and halogenated compounds [50,51]. Ninety genes were detected in the genome of strain AQ5-07 that are involved in the metabolism of aromatic compounds, including metabolism of central aromatic intermediates such as salicylate, protocatechuate, and catechol. Similar results have been reported in *R. erythropolis* strain CCM2595 [52], *Rhodococcus* sp. strain RHA1 [53], and *Rhodococcus* sp. strain 311R [40]. For instance, the genome of *R. erythropolis* strain CCM2595 harbours the *catRABC* cluster coding for enzymes involved in catechol degradation [52]. Nahar et al. [43] similarly reported that genes for aromatic compound metabolism are present in the genomes of three sub-Antarctic *Rhodococcus* spp.

Tomás-Gallardo et al. [54] noted that a number of transcriptional regulators have been identified in *Rhodococcus* genome sequences, with most belonging to LysR and IclR-type families. Several transcriptional regulators of *Rhodococcus* spp. involved in regulating the degradation of aromatic compounds have been characterised in previous studies [51,55,56,57]. Transcriptional regulator CatR is transcribed in the opposite direction to *catABC* genes, coding for the IclR family transcriptional regulator. The amino acid sequence of CatR (accession number RAL35564) from strain AQ5-07 shared 99.6% and 98.8% similarity with CatR from *R. erythropolis* PR4 (accession number BAH36132) and *R. erythropolis* CCM2595 (accession number AGT94972), respectively. A similar result was reported in *Rhodococcus opacus* 1CP, where the CatR gene encoded a regulator belonging to the IclR family [57,58]. Previous studies have shown that CatR is responsible for the transcriptional control of the *cat* operon [59,60]. For example, Cámara et al. [61] noted that most members of the genus *Pseudomonas* express CatRBCA gene cluster controlled by CatR in response to the substrate muconate. Eulberg and Schlömann [58] demonstrated that CatR regulates the *catABC* expression in *R. opacus* 1CP by binding the protein within the intergenic *catR*-*catA* region. Tropel and van der Meer [62] noted that IclR-type regulators generally function as repressors, although IclR may also function as activators.

The *pheR*-*pheA2A1* operon was detected contiguous to the *cat* operon. A similar organisation of this gene cluster has been reported in *R. erythropolis* CCM2595, where the gene cluster including *catA*, *catB*, *catC*, *catR*, *pheR*, *pheA2*, *pheA1* was involved in the *ortho*-cleavage pathway of phenol [56]. The PheR gene is transcribed in the opposite direction to *pheA2* and *pheA1* (Figure 5), which is an AraC-type transcriptional regulator. Szőköl et al. [56] suggested that PheR activates the *pheA2* promoter of *R. erythropolis* CCM2595. The amino acid sequence of PheR shared 99.0% similarity with PheR from *R. erythropolis* (accession no. CAJ01323). The *pheA2A1* genes coding for a two-component phenol hydroxylase are involved in the first step of phenol degradation [56]. The amino acid sequences of PheA2 and PheA1 obtained here were respectively 100% and 99.8% similar to PheA1 (accession no. ABS30825.1) and PheA2 (accession no. WP_019745631) from *R. erythropolis.* The two components of phenol hydroxylase encoded by PheA1 and PheA2 have been reported in several *Rhodococcus* spp., including *R. opacus* 1CP [63], *R. erythropolis* UPV-1 [64], *R. erythropolis* CCM2595, and *R. jostii* RHA1 [56].

Catechol 1, 2-dioxygenase, muconate cycloisomerase, and muconolactone isomerase, the first three enzymes involved in catechol degradation, were identified as coded by the *catABC* operon. Previous studies have reported *catRABC* genes in *Rhodococcus* sp. strain RHA1 [53] and *R. erythropolis* CCM2595 [57]. The amino acid sequence of CatA shared the highest identity (99.6%) with CatA from *R. erythropolis* strain PR4 (accession no. WP_019745628). Many studies have shown that certain aerobic bacteria metabolise aromatic compounds to non-toxic intermediates of the TCA cycle via the *ortho*-pathway using C12D or the *meta*-pathway using C23D. As bioinformatics analyses of the whole genome sequence of strain AQ5-07 did not identify the gene encoding C23D and no C23D activity was detectable from cell-free crude extract, it is likely that, in strain AQ5-07, catechol is first converted to CCMA by the C12D enzyme encoded by *catA*, which is then metabolised by CatB and CatC.

A number of phenol-degrading microorganisms have been studied, and the pathways for the aerobic phenol degradation are now firmly established; nonetheless, phenol degradation pathways based on genetic traits in Antarctic bacteria have yet to be thoroughly explored [65]. It has been established that members of the genus *Rhodococcus* play important roles in the biodegradation of compounds that cannot be easily transformed by other organisms [51]. Moreover, several psychrotolerant *Rhodococcus* spp. isolated from Antarctica are capable of degrading hydrocarbons under low temperatures [65,66,67,68]. *Rhodococcus* sp. AQ5-07 isolated from Antarctic soil is a cold-adapted strain that is able to degrade phenol at optimum temperatures of 10–15 °C [30]. The results of genomic analyses of strain AQ5-07 revealed the presence of genes for the complete enzyme system (PheA1A2, C12D, muconate cycloisomerase, muconolactone isomerase, 3-oxoadipate enol-lactonase, 3-oxoadipate CoA-transferase, acetyl-coa acyltransferase) that can completely metabolise phenol to products entering the TCA cycle via the β-ketoadipate (*ortho*-) pathway [57]. However, the presence of these genes does not directly confirm each enzymes’ functionality, and further investigation is required using other approaches.

The pathway of phenol degradation of strain AQ5-07 was determined as the *ortho*-pathway based on the absence of C23D and the confirmation of C12D activity with specific enzyme activity of 23.603 U/mg after 10 min of incubation. In comparison, a higher C12D activity of 1730 U/mg was reported in *Rhodococcus* sp. strain RHA1 [53]. A number of other studies have also identified and measured C12D activities in *Rhodococcus* spp. [69,70,71]. A high C23D activity in *Rhodococcus ruber* UKMP-5M in the early minutes of incubation was also reported by Tavakoli and Hamzah [72].

## 4. Materials and Methods

### 4.1. Strain and Phenol Medium

*Rhodococcus* sp. strain AQ5-07 was isolated from Antarctic soil obtained on 9 September 2007 from King George Island, South Shetland Islands (62°09′7.2″ S, 58°11.4″ W) [30]. Representative material is deposited in the Microbial Culture Collection Unit (UNiCC) of Universiti Putra Malaysia under reference number “UPMC 1202”.

Phenol medium (0.5 g/L) was prepared in 1 L volume by adding 0.4 g KH_2_PO_4_, 0.2 g K_2_HPO_4_, 0.1 g MgSO_4_, 0.1 g NaCl, 0.01 g MnSO_4_.H_2_O, 0.01 g Fe_2_(SO_4_)_3_.H_2_O, 0.01 g Na_2_MoO_4_.H_2_O, and 0.4 g (NH_4_)_2_SO_4_ to distilled water. The medium was adjusted to pH 7.2 using NaOH, checked with a pH meter (Mettler Toledo FiveEasy Plus™, Greifensee, Switzerland). The medium was autoclaved for 15 min at 121 °C. The sterilised medium was then augmented with 0.5 g crystalline phenol.

### 4.2. Analytical Procedure

Bacterial growth was determined by measuring Optical Density 600 using a U.V Mini 1240 Shimadzu Spectrophotometer at a wavelength of 600 nm. Meanwhile, the determination of phenol concentration was carried out using a UV-vis spectrophotometric method with 4-aminoantipyrine as colorimetric agent at a wavelength of 510 nm [73].

### 4.3. Optimisation Using Statistical Approach

In a preliminary experiment, the effects of different parameters were studied as single variables in OFAT without considering interactions between the variables. Experimental optimisation using a statistical approach was conducted subsequent to OFAT to effectively determine the optimum levels including interactions among variables. In the statistically designed experiment, the range of each parameter was chosen based on the results of OFAT [30]. The experiments were performed in triplicate, and assessments of phenol degradation were made after an incubation period of 84 h for AQ5-07.

#### 4.3.1. Plackett–Burman Design

The Plackett–Burman design was employed to screen for significant factors prior to statistical optimization with RSM. The statistically planned experiments were designed and analysed by using statistical software Design-Expert version 6 (Stat-Ease Inc., Minneapolis, MN, USA). The experimental ranges of each parameter were selected based on the results from OFAT. Four important factors as previously identified in OFAT were optimized and screened at two levels (−1 and 1) using the Plackett–Burman design. Each statistically planned experiment was conducted in triplicate, and the significance of the effect of each factor on phenol degradation was determined. The experimental ranges and levels of the four independent variables tested in the Plackett–Burman design for strain AQ5-07 are shown in Table 7. The analysis shows a total number of 12 experimental designs, with each row of the table consisting of four independent variables for the selected strain, where A is the temperature, B is pH, C is the concentration of NaCl (g/L), and D is the concentration of the nitrogen source (g/L).

#### 4.3.2. CCD

The significant variables (*p* < 0.05) identified through Plackett–Burman were optimised by CCD. Design-Expert version 6 software was used to design the statistical experiments of CCD. The selected significant variables were analysed at five different levels (−2, −1, 0, 1, 2) for AQ5-07, as shown in Table 8. A total of 20 runs were designed for AQ5-07 using the three significant variables.

The quadratic model of CCD was used to describe the relationship between response and independent variables based on a second-order polynomial equation as follows:(1)$$Y=\beta_{0}\overset{k}{\underset{i=1}{\sum }}\beta_{i}X_{i\;}+\;\overset{k}{\underset{i=1}{\sum }}\beta_{ii}X_{i}^{2}+\overset{k}{\underset{1\leq i\leq j}{\sum }}\beta_{ij}X_{i\;}X_{j}$$
where *Y* is the phenol degradation (response); *X_i_* and *X_j_* are the independent variables; *k* is the number of variables; $\beta_{0}$ is the model intercept; $\beta_{i}$ is the ith linear coefficient; $\beta_{ii}$ is the ith quadratic coefficient; and $\beta_{ij}$ is the *ij*th interaction coefficient [33]. Analysis of variance (ANOVA) was used to determine the significance of the model and regression coefficients. The fit of the model was evaluated by the determination coefficient (R^2^), and statistical significance of the model was determined by Fisher’s *F*-test.

#### 4.3.3. Validation of Experiments

Based on the results from CCD, the predicted value of response was generated using Design-Expert version 6 (Stat-Ease Inc., Minneapolis, MN, USA) to permit the validation of experiments with the values of significant factors given. Independent statistically designed experiments were carried out in triplicate to validate the predicted model. Subsequently, the actual value (percentage of phenol degradation) obtained from the experiment was compared with the predicted value of response generated by CCD [74].

### 4.4. Whole Genome Sequencing

#### 4.4.1. Extraction of Genomic DNA

The selected bacterial strain was cultured in 50 mL of nutrient broth (Merck) on a shaking incubator at 15 °C for 48 h. The cultured broth (1.5 mL) was transferred to an Eppendorf tube and centrifuged at 15,000× *g* for 2 min in order to remove the supernatant. Subsequently, the genomic DNA was extracted using the GeneJET Genomic DNA Extraction Kit (Thermo Scientific, Waltham, MA, USA) following the manufacturer’s protocol. The extracted DNA was examined using 1.0% (*w*/*v*) agarose gel electrophoresis stained with 0.5 µg/mL of ethidium bromide (Vivantis Technologies Sdn Bhd, Subang Jaya, Malaysia) at 100 V for 40 min. Gel electrophoresis was performed using the Lambda/Hind III marker (Vivantis Technologies Sdn Bhd, Subang Jaya, Malaysia) as ladder to examine the size of the extracted genomic DNA prior to observation of the gel under UV light. Subsequently, the purity and the concentration of extracted genomic DNA were assessed using a Nanodrop spectrophotometer (Bio-Rad, Des Plaines, IL, USA). The extracted genomic DNA of strain AQ5-07 was sent to Beijing Novogene Bioinformatics Technology Co., Ltd. for whole genome sequencing.

#### 4.4.2. Genome Sequencing and Assembly

Whole genome sequencing of strain AQ5-07 was performed on the Illumina HiSeq 2500-PE125 platform with massively parallel sequencing Illumina technology. Prior to sequencing, the harvested genomic DNA was detected using sodium dodecyl sulfate polyacrylamide gel electrophoresis (SDS-PAGE) and quantified using Qubit, A-tailed, ligated to paired-end adaptors, and PCR amplified with a 500 bp insert, and a mate-pair library with an insert size of 5 kb was used for library construction at Beijing Novogene Bioinformatics Technology Co. Ltd. Filtration of Illumina PCR adapter reads and low quality reads from the paired-end and mate-pair library was performed for quality control. SOAPdenovo was used to assemble all the paired reads into a number of scaffolds followed by handling of the filter reads by the next step of gap-closing [75].

#### 4.4.3. Gene Prediction and Annotation

The GeneMarkS program [76] was used to retrieve the related coding genes of strain AQ5-07. The interspersed repetitive sequences were predicted using RepeatMasker [77]. Tandem repeats were analysed by TRF (tandem repeats finder) [78]. Transfer RNA (tRNA) genes were predicted by the tRNAscan-SE [79]. Ribosomal RNA (rRNA) genes were analysed using the rRNAmmer [80]. Small nuclear RNAs (snRNA) were predicted by Basic Local Alignment Search Tool (BLAST) against the Rfam database [81]. Annotations of the whole genome sequences were performed using the automated web-based tool, Rapid Annotations using subsystems Technology (RAST) server, with the SEED database [82]. To screen for phenol degradative genes, the amino acid sequences were searched against the protein sequence database from the BLAST [83], Uniprot [84], the Kyoto Encyclopedia of Genes and Genomes (KEGG) [85], and Clusters of Orthologous Groups (COG) databases [86]. The Whole Genome Shotgun project has been deposited at DNA Data Bank of Japan/ European Nucleotide Archive/GenBank (DDBJ/ENA/Genbank) under accession QEUH00000000.

### 4.5. Enzyme Assays

#### 4.5.1. Preparation of Cell Extracts

Phenol-degrading bacterial strains were cultured in 10 mL of minimal salt medium (MSM) containing 0.7 g/L phenol on a shaking incubator at 150 rpm at 15 °C for 5 d. Cells (2 mL) were then harvested by centrifugation at 4500× *g* for 15 min. This was followed by washing with 50 mM of phosphate buffer at pH 7.5 and resuspension in 2 mL of the same buffer. Cells were then disrupted by sonication with 30 s intervals of sonication and 30 s intervals of interruption for a total of 6 min in an ice-cooled bath [87]. Pellets were removed by centrifugation at 9000× *g* for 30 min at 4 °C. The collected supernatants were used in the following enzyme assays [88,89].

#### 4.5.2. Enzyme Assay of catechol 1, 2 dioxygenase (C12D)

Enzyme activity of C12D was determined by confirming the formation of cis, cis–muconic acid (CCMA) in the presence of catechol as substrate. In the C12D enzyme assay, cell-free extract (20 µL) was added to 50 mM phosphate buffer (pH 7.5) containing 20 mM Na_2_EDTA and 50 mM catechol to give a final volume of 1 mL [90]. Cis, cis-muconic acid formation was measured spectrophotometrically at 260 nm for 1 h at 10 min intervals in an ice bath. The crude extracts were pre-treated for 5 min with 0.01% (*v*/*v*) H_2_O_2_ to suppress any activity of C23D prior to the enzyme assays [91,92]. The absorbance of each sample was read in a quartz cuvette, and distilled water was used as a blank. The control was prepared by replacing the cell-free extracts with distilled water. The extinction coefficient of CCMA was determined as $\varepsilon_{260\;nm}$ = 16,800/M cm [88,90]. One unit of enzyme activity was defined as the amount of enzyme required to generate 1 µmol of CCMA per minute.

#### 4.5.3. Enzyme Assay of catechol 2, 3 dioxygenase (C23D)

Enzyme activity of C23D was determined spectrophotometrically at 375 nm based on the method developed by Hupert-Kocurek et al. [90]. The formation of 2-hydroxymuconic semialdehyde (2-HMS) was measured at 375 nm at 10 min intervals for 1 h in an ice bath. Cell-free extract (20 µL) was added to 980 µL of 50 mM phosphate buffer (pH 7.5) containing 50 mM catechol to give a final volume of 1 mL. The absorbance of each sample was read in a quartz cuvette, and distilled water was used as a blank. The control was prepared by replacing the cell-free extracts with distilled water. The extinction coefficient of 2-HMS was measured as $\varepsilon_{375\;\mathrm{nm}}$ = 36,000/M cm [90,93]. One unit of enzyme activity was defined as the amount of enzyme required to generate 1 µmol of 2-HMS per minute.

## 5. Conclusions

Statistical optimisation using RSM resulted in faster phenol degradation compared to OFAT analysis in *Rhodococcus* sp. strain AQ5-07. The draft genome sequence of strain AQ5-07 was documented, and analysis of the genome revealed a complete enzyme system of the β-ketoadipate (ortho-) pathway of phenol degradation, initiated by a two-component phenol hydroxylase PheA1A2. Enzyme assays confirmed activity of only one catechol ring cleavage enzyme, C12D, consistent with the results from whole genome sequencing. The availability of the whole genome sequence of strain AQ5-07 provides a means for extending understanding of the physiology, the evolution, and the functions of this indigenous Antarctic phenol-degrading bacterium, including its future potential role in bioremediation. Future research will focus on the analysis of gene functions, regulation mechanisms, and cold-adaptation characteristics based on information contained in the whole genome sequence of strain AQ5-07.

## Figures and Tables

**Figure 1 ijms-21-09363-f001:**
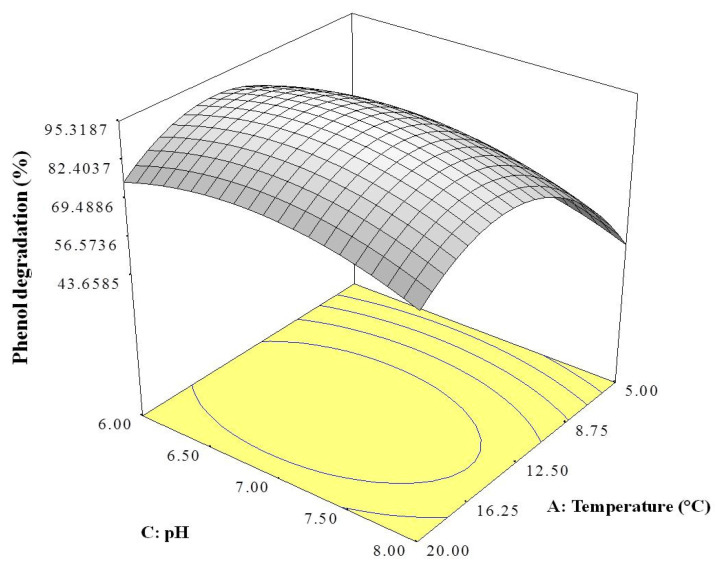
3D response surface plot showing the interaction effect of temperature and pH on phenol degradation by strain AQ5-07.

**Figure 2 ijms-21-09363-f002:**
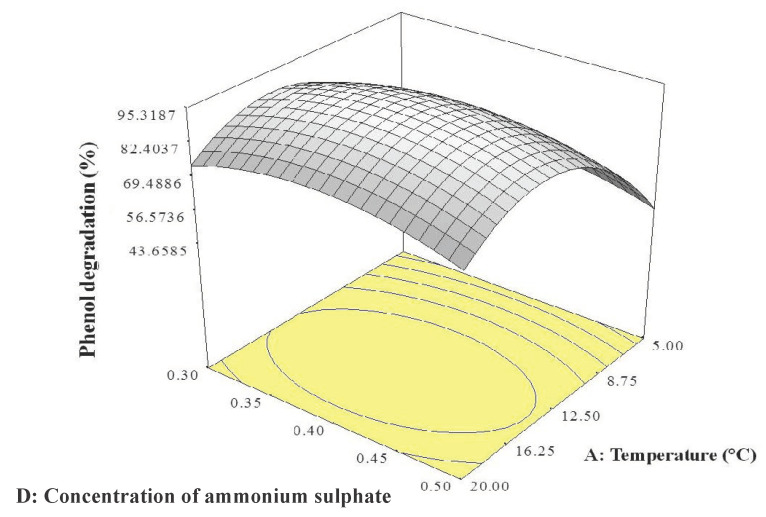
3D response surface plot showing the interaction effects of temperature and concentration of ammonium sulphate on phenol degradation by strain AQ5-07.

**Figure 3 ijms-21-09363-f003:**
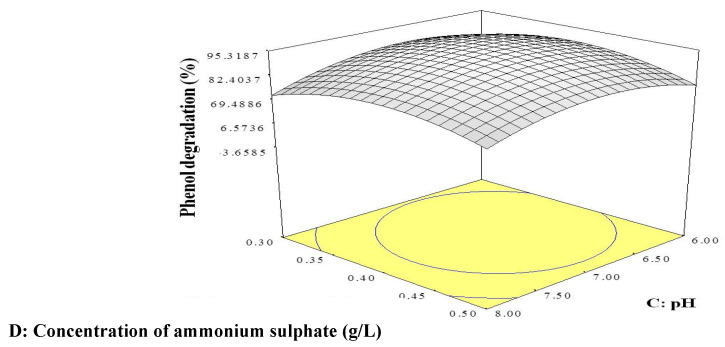
3D response surface plot showing the interaction effects of pH and concentration of ammonium sulphate on phenol degradation by strain AQ5-07.

**Figure 4 ijms-21-09363-f004:**
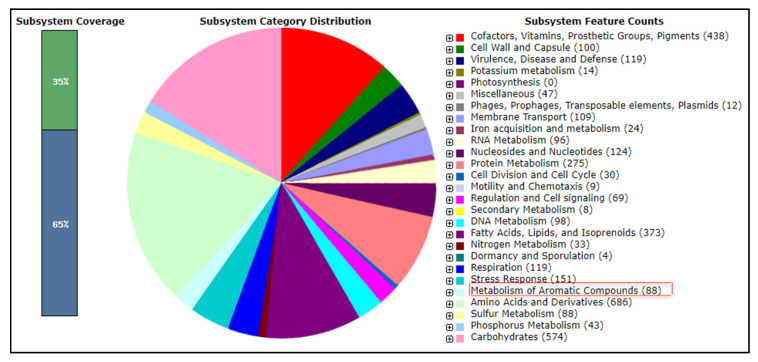
Subsystem category distribution for *Rhodococcus* sp. strain AQ5-07 annotated using the Rapid Annotations using Subsystems Technology (RAST) server. The red box indicates the number of genes involved in the metabolism of aromatic compounds.

**Figure 5 ijms-21-09363-f005:**
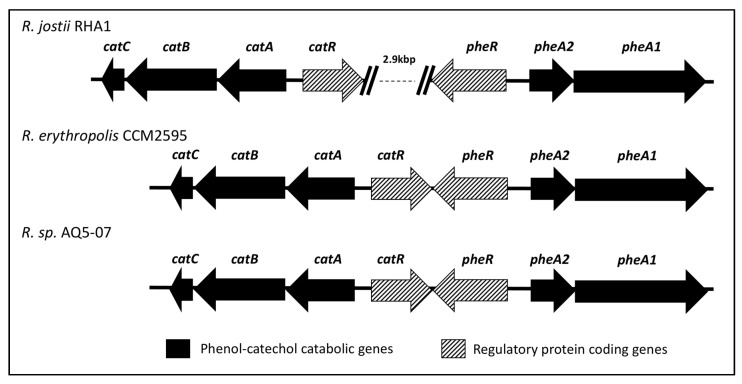
Physical map of the gene cluster containing phenol hydroxylase and catechol 1, 2-dioxygenase genes from *Rhodococcus* sp. strain AQ5-07 in comparison to *Rhodococcus jostii* RHA1 and *Rhodococcus erythropolis* CCM2595. The putative functions of the genes in the cluster were predicted as follows: muconolactone isomerase, CatC; muconate cycloisomerase, CatB; catechol 1,2-dioxygenase, CatA; IclR family transcriptional regulator, CatR; AraC family transcriptional regulator, PheR; phenol hydroxylase small subunit A2, PheA2; phenol hydroxylase large subunit A1, PheA1.

**Figure 6 ijms-21-09363-f006:**
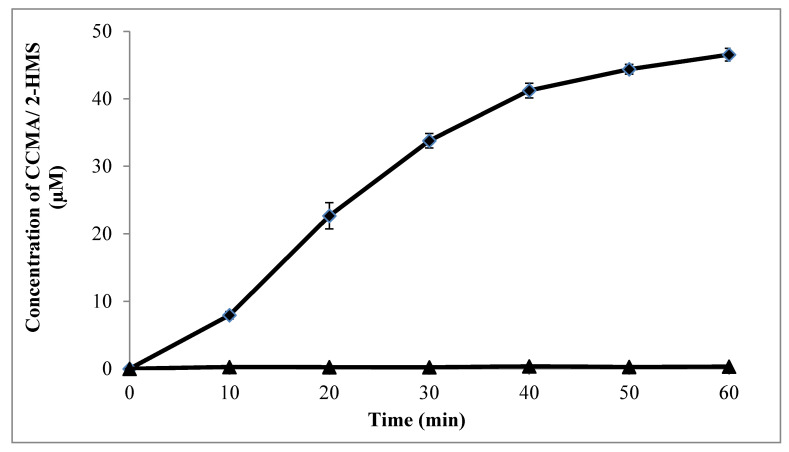
Formation of cis, cis-muconic acid (filled diamond), and 2-hydroxymuconic semialdehyde (filled triangle) by catechol dioxygenases of *Rhodococcus* sp. strain AQ5-07. Error bars represent mean ± standard deviation for three replicates.

**Table 1 ijms-21-09363-t001:** Plackett–Burman experimental design matrix with phenol degradation by strain AQ5-07.

Run	A	B	C	D	Phenol Degradation (%)
1	25	0.30	8	0.1	53.04
2	5	0.05	8	0.5	49.31
3	5	0.30	6	0.1	49.08
4	25	0.30	6	0.5	61.08
5	5	0.30	8	0.1	44.12
6	5	0.05	6	0.5	50.34
7	25	0.05	8	0.5	56.64
8	5	0.05	6	0.1	46.43
9	25	0.30	6	0.5	59.36
10	25	0.05	8	0.1	52.68
11	25	0.05	6	0.1	56.52
12	5	0.30	8	0.5	44.58

A: temperature; B: concentration of NaCl; C: pH; D: concentration of (NH_4_)_2_SO_4_.

**Table 2 ijms-21-09363-t002:** Analysis of variance (ANOVA) for phenol degradation by strain AQ5-07 with Plackett–Burman design.

Source	Sum of Squares	Degree of Freedom	Mean Square	F Value	Prob > F
Model	344.5220	8	43.0653	57.7867	0.0034 **
A	129.9352	1	129.9352	174.3524	0.0009 ***
B	3.0380	1	3.0380	4.0765	0.1368
C	37.8766	1	37.8766	50.8245	0.0057 **
D	11.7988	1	11.7988	15.8322	0.0284 *
AC	2.5548	1	2.5548	3.4281	0.1612
AD	0.0870	1	0.0870	0.1168	0.7551
BC	5.6012	1	5.6012	7.5159	0.0712
BD	7.2682	1	7.2682	9.7528	0.0524
Residual	2.2357	3	0.7452		
Lack of Fit	0.7565	2	0.3783	0.2557	0.8134
Pure Error	1.4792	1	1.4792		
Cor Total	346.7578	11			
Std dev	0.86	R^2^	0.9936		
Mean	51.93	Adjusted R^2^	0.9764		
C.V	1.66	Predicted R^2^	0.9290		
PRESS	24.63	Adeq Precision	21.1694		

A: temperature; B: concentration of NaCl; C: pH; D: concentration of (NH_4_)_2_SO_4_; *****
*p* < 0.05, ** *p* < 0.01, *** *p* < 0.001.

**Table 3 ijms-21-09363-t003:** Central composition design (CCD) experimental matrix with corresponding experimental and predicted values of phenol degradation for strain AQ5-07.

Run Order	A	C	D	Phenol Degradation (%)	
				Experimental Value	Predicted Value
1	12.5	7.0	0.40	93.91	93.45
2	12.5	7.0	0.40	94.10	93.45
3	12.5	7.0	0.23	69.72	69.42
4	25.0	7.0	0.40	47.40	46.21
5	20.0	6.0	0.50	63.48	64.11
6	12.5	7.0	0.40	92.53	93.45
7	20.0	8.0	0.30	60.60	61.10
8	12.5	5.3	0.40	73.56	73.10
9	12.5	8.7	0.40	62.64	62.60
10	5.0	8.0	0.30	33.96	33.69
11	12.5	7.0	0.57	67.20	67.00
12	12.5	7.0	0.40	93.52	93.45
13	12.5	7.0	0.40	93.72	93.45
14	5.0	6.0	0.30	43.32	43.23
15	0	7.0	0.40	2.76	3.43
16	20.0	6.0	0.30	68.76	69.44
17	12.5	7.0	0.40	92.82	93.45
18	5.0	8.0	0.50	36.48	36.16
19	20.0	8.0	0.50	60.84	61.30
20	5.0	6.0	0.50	40.32	40.18

A: temperature; C: pH; D: concentration of (NH_4_)_2_SO_4_.

**Table 4 ijms-21-09363-t004:** Analysis of variance (ANOVA) for phenol degradation by strain AQ5-07 with CCD.

Source	Sum of Squares	DF	Mean Square	F Value	Prob > F
Model	12,042.1628	9	1338.0181	2358.7788	<0.0001 ***
A	2233.4754	1	2233.4754	3937.3716	<0.0001 ***
C	131.4727	1	131.4727	231.7720	<0.0001 ***
D	6.9752	1	6.9752	12.2965	0.0057 **
A^2^	8534.5562	1	8534.5562	15,045.4843	<0.0001 ***
C^2^	1169.5896	1	1169.5896	2061.8578	<0.0001 ***
D^2^	1136.9217	1	1136.9217	2004.2680	<0.0001 ***
AC	0.7200	1	0.7200	1.2693	0.2862
AD	2.5992	1	2.5992	4.5821	0.0580
CD	15.2352	1	15.2352	26.8580	0.0004 ***
Residual	5.6725	10	0.5673		
Lack of Fit	3.7190	5	0.7438	1.9037	0.2484
Pure Error	1.9535	5	0.3907		
Cor Total	12,047.8353	19			
Std dev	0.75	R^2^	0.9995		
Mean	64.58	Adjusted R^2^	0.9991		
Coefficient of Variance	1.17	Predicted R^2^	0.9975		
PRESS	30.56	Adeq Precision	169.0312		

A: temperature; C: pH; D: concentration of (NH_4_)_2_SO_4_; ** *p* < 0.01, *** *p* < 0.001.

**Table 5 ijms-21-09363-t005:** Genomic features of *Rhodococcus* sp. strain AQ5-07.

Feature	Count/Value
Genome size (bp)	6,749,221
GC content (%)	62.4
Number of contigs	34
Length of the longest contig (bp)	1,378,316
Number of Subsystems	436
Number of coding sequences (CDSs)	6545
Number of rRNAs	4
Number of tRNAs	75

**Table 6 ijms-21-09363-t006:** Gene candidates involved in complete phenol catabolism identified in the genome of *Rhodococcus* sp. AQ5-07.

Gene Name (Locus Tag)	Gene Products	Accession No.	Amino Acid Residues (Aa)	COG No.	KEGG No.
*pheA1* (S2GM001986)	Phenol hydroxylase large subunit A1	RAL35045	542	COG2368	K03380
*pheA2* (S2GM001985)	Phenol hydroxylase small subunit A2	RAL35044	189	COG1853	K03380
*catA* (S2GM001982)	Catechol 1,2-dioxygenase	RAL35043	279	COG3485	K03381
*catB* (S2GM001981)	Muconate cycloisomerase	RAL35564	373	COG4948	K01856
*catC* (S2GM001980)	Muconolactone isomerase	RAL35042	93	COG4829	K03464
*praD* (S2GM002436)	3-oxoadipate enol-lactonase	RAL34922	270	COG0596	K01055
*pcaI* (S2GM003023)	3-oxoadipate coa-transferase, alpha subunit	RAL34191	303	COG1788	K01031
*pcaJ* (S2GM003024)	3-oxoadipate coa-transferase, beta subunit	RAL34192	261	COG2057	K01032
*fadA/ fadI* (S2GM005247)	Acetyl-coa acyltransferase	RAL31833	411	COG0183	K00632

**Table 7 ijms-21-09363-t007:** Experimental range and level of independent variables tested in Plackett–Burman design for *Rhodococcus* sp. AQ5-07.

Variables	Symbol	Unit	Experimental Value	
			Low (−1)	High (+1)
Temperature	A	°C	5	25
Concentration of NaCl	B	g/L	0.05	0.25
pH	C	-	6	8
Concentration of (NH_4_)_2_SO_4_	D	g/L	0.1	0.5

**Table 8 ijms-21-09363-t008:** Experimental range and level of independent significant variables tested in CCD for *Rhodococcus* sp. AQ5-07.

Variables	Symbol	Unit	Experimental Value				
			−2	−1	0	+1	+2
Temperature	A	°C	0	5.0	10.0	20.0	25.0
pH	C	-	5.3	6.0	7.0	8.0	8.7
Concentration of (NH_4_)_2_SO_4_	D	g/L	0.23	0.30	0.40	0.50	0.57

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
