# Peer review of "Statistical Optimisation of Phenol Degradation and Pathway Identification through Whole Genome Sequencing of the Cold-Adapted Antarctic Bacterium, Rhodococcus sp. Strain AQ5-07"

_ijms, 2020, doi:10.3390/ijms21249363_

Round 1
Reviewer 1 Report
The manuscript describes degradation of phenol by cold-adapted Rhodococcus. The authors also shortly describe some properties of the strain derived from the complete genome sequence.
The manuscript brings some new aspects of phenol degradation at low temperatures. The manuscript deserves publication after careful addressing several concerns.
Major comments:
1) The authors derived most of the presented data concerning genes involved in the phenol degradation just from the genome sequence. Evidence about the activity (transcription, expression) of the genes is missing. Activity of a single enzyme from the supposed degradation pathway was only determined. It should be explained why the activity of phenol hydroxylase (the first enzyme of the pathway) was not determined. The genes are called as "candidates" for the genes responsible for phenol degradation. It is not clear, if the authors consider the detected and described gene cluster really responsible for the phenol degradation. If they delete e.g. the pheA2 or pheA1 gene, they could be sure about the participation of these genes in the degradation. At the same time, they could be sure that this is really the only gene/enzyme responsible for the first reaction of the degradation pathway.
2) At line 186, there is a mysterious mention that "redundancy of metabolic pathways and genes in this genus identified in the analysis of the strain AQ5-07 genome complicates elucidation of the biodegradation pathways." It should be clearly stated if some other genes (than in the described cluster) potentially involved in phenol degradation were detected in the AQ5-07 genome. For example, the strain R. jostii RHA1 contains two clusters of phe/cat genes.
3) It would be interesting if the protein sequences of the phenol degradation enzymes are analyzed with the focus of characteristics of "cold-adapted proteins/enzymes". Some psychrophiles possess several enzymes with the same activity, but with different temperature optimum. These enzymes differ by their amino-acid content and can be defined as adapted to cold or "normal" temperatures.
4) The Rhodococcus cells were cultivated in minimal medium with phenol as a sole carbon and energy source. The readers would like to see the comparison of the growth curves with glucose (or other good substrate) and with phenol only. It is not clear if the phenol degrading enzymes were induced by phenol. Did inoculum for phenol degradation contain phenol? Was there a difference in growth of cells adapted to phenol and cells for which the substrate was abruptly replaced by phenol?
5) The assay of catechol dioxygenase was done with cells harvested after 5-d cultivation. Why? Was the activity measured with cells from other time points? The enzyme activity was measured in ice bath, that is at 0 °C. Why? I would expect measuring at the same temperature as the degradation experiments were performed. Did you try to measure the enzyme activity at different temperatures? 10, 20, 30 °C? The assay temperature (cultivation, measurement of enzyme activity) should be mentioned in legend to Fig. 7.
6) I would move the Conclusion just next to the end of Discussion. The Conclusion should be rewritten. It is now just summarizing some facts found in the study. Moreover, it only mentions the genome sequencing, although it seems that the emphasis of the manuscript is given mainly to phenol degradation (Results, lines 92-155 + 4 tables + 3 figures). Some general ideas particularly concerning degradation at low temperatures and the properties of the Rhodococcus strain should be presented. There should be some outlook, what the authors consider most interesting to study next. Since the manuscript contains two different parts (biotechnological and genetic), there should be some balance in information. Moreover, advantages of such combined approach in a single paper should be explained.
7) There are many stylistic errors and clumsy expression and sentences in the manuscript. The text should be checked and corrected by a professional English editing/proofreading service. The paper would look much better then.
8) The title should be adapted. Statistical approach to what? Metabolic pathway is not in the center of the manuscript since it was just derived from the genome sequence. The title should aptly express real content of paper.
9) If the whole AQ5-07 genome is available, it should be stated which Rhodococcus species and strains seem the closest from the genetic point of view.
Minor comments:
1) Line 32: RSM – write in full
2) L 39: "we predicted the degradation pathway" - repetition
3) L 41: "improved understanding of the mechanism of phenol degradation” – this seems exaggerated. No mechanisms were studied (transcription, regulation, induction).
4) L 63: Are you sure that phenol is still used as disinfectant? Please, provide citation.
5) L 89: "involvement of key enzymes" – I could find a proof of the presence of only a single enzyme activity.
6) L 166: please, define "subsystem". I would call it e.g. "functional gene group" of "gene category"
7) L 169: "subsystem of carbohydrates.. cofactors.." I would call it "carbohydrate metabolism, cofactor synthesis"
8) L 170: antifreeze protein, cold shock proteins – please provide some comparisons with genes and enzymes from other psychrotrophic bacteria. This seems to be an interesting point.
9) L 179: "The results obtained" Which results?
10) Figure 4: Could you provide some more data about the detected regulators and 2-component systems? Did you find what is a number of potential sigma factors? These facts deserve a short subsection.
11) Figure 5: It would be better to put the gene or enzyme names above the arrows
12) Figure 6, L 217: "… presence of enzyme-encoding genes"
13) L 262: "related to metabolism of amino acids"
14) L 295-297: Please, rephrase
15) Optimum cultivation conditions should be mentioned in Introduction, if there was a paper describing them
16) L 459: I guess that catechol was a substrate rather than an inducer
Author Response
Major comments:
Comment no. 1
The authors derived most of the presented data concerning genes involved in the phenol degradation just from the genome sequence. Evidence about the activity (transcription, expression) of the genes is missing. Activity of a single enzyme from the supposed degradation pathway was only determined. It should be explained why the activity of phenol hydroxylase (the first enzyme of the pathway) was not determined. The genes are called as "candidates" for the genes responsible for phenol degradation. It is not clear, if the authors consider the detected and described gene cluster really responsible for the phenol degradation. If they delete e.g. the pheA2 or pheA1 gene, they could be sure about the participation of these genes in the degradation. At the same time, they could be sure that this is really the only gene/enzyme responsible for the first reaction of the degradation pathway.
Answer: We have removed all statements indicating the detected genes are directly responsible for the phenol degradation activity. We agree further investigation works are needed to confirm this.
Comment no. 2
At line 186, there is a mysterious mention that "redundancy of metabolic pathways and genes in this genus identified in the analysis of the strain AQ5-07 genome complicates elucidation of the biodegradation pathways." It should be clearly stated if some other genes (than in the described cluster) potentially involved in phenol degradation were detected in the AQ5-07 genome. For example, the strain R. jostii RHA1 contains two clusters of phe/cat genes.
Answer: This statement has been removed.
Comment no. 3
It would be interesting if the protein sequences of the phenol degradation enzymes are analyzed with the focus of characteristics of "cold-adapted proteins/enzymes". Some psychrophiles possess several enzymes with the same activity, but with different temperature optimum. These enzymes differ by their amino-acid content and can be defined as adapted to cold or "normal" temperatures.
Answer: Yes, this is very interesting. Thank you for this suggestion. We did some preliminary analyses to find molecular evidence of cold-adapted proteins (calculating specific amino acid residues), however, we have yet to see appreciable differences. We will continue looking in to this.
Comment no. 4
The Rhodococcus cells were cultivated in minimal medium with phenol as a sole carbon and energy source. The readers would like to see the comparison of the growth curves with glucose (or other good substrate) and with phenol only. It is not clear if the phenol degrading enzymes were induced by phenol. Did inoculum for phenol degradation contain phenol? Was there a difference in growth of cells adapted to phenol and cells for which the substrate was abruptly replaced by phenol?
Answer: We did growth pattern studies and reported it previously [30], however, we have not yet tried any comparison studies using substrates other than phenol. We think it would be interesting to be able to compare these aspects with other well studied Rhodococcus sp. We will consider doing this study in the future.
Comment no. 5
The assay of catechol dioxygenase was done with cells harvested after 5-d cultivation. Why? Was the activity measured with cells from other time points? The enzyme activity was measured in ice bath, that is at 0 °C. Why? I would expect measuring at the same temperature as the degradation experiments were performed. Did you try to measure the enzyme activity at different temperatures? 10, 20, 30 °C? The assay temperature (cultivation, measurement of enzyme activity) should be mentioned in legend to Fig. 7.
Answer: The cell-free crude extract samples used for this assay were prepared by sonication. To prevent enzyme from denaturing due to heat from this process, cells were sonicated on ice. We were also concerned about losing enzymes’ viability after cell disruption, hence the assay was conducted at low temperature. We regard putting on ice is closer to 4°C rather than 0°C. We did not conduct this assay at any other temperatures.
Comment no. 6
I would move the Conclusion just next to the end of Discussion. The Conclusion should be rewritten. It is now just summarizing some facts found in the study. Moreover, it only mentions the genome sequencing, although it seems that the emphasis of the manuscript is given mainly to phenol degradation (Results, lines 92-155 + 4 tables + 3 figures). Some general ideas particularly concerning degradation at low temperatures and the properties of the Rhodococcus strain should be presented. There should be some outlook, what the authors consider most interesting to study next. Since the manuscript contains two different parts (biotechnological and genetic), there should be some balance in information. Moreover, advantages of such combined approach in a single paper should be explained.
Answer: Conclusion statements updated.
Comment no. 7
There are many stylistic errors and clumsy expression and sentences in the manuscript. The text should be checked and corrected by a professional English editing/proofreading service. The paper would look much better then.
Answer: This manuscript has be proofread by a native English speaker in our research team.
Comment no. 8
The title should be adapted. Statistical approach to what? Metabolic pathway is not in the center of the manuscript since it was just derived from the genome sequence. The title should aptly express real content of paper.
Answer: Title was changed to reflect the revised manuscript
Comment no. 9
If the whole AQ5-07 genome is available, it should be stated which Rhodococcus species and strains seem the closest from the genetic point of view.
Answer: Yes, strain AQ5-07 genome is available. For detailed genomic study report, we are currently preparing a manuscript for future publication.
Minor comments:
Comment no. 1
Line 32: RSM – write in full
Answer: Corrected
Comment no. 2
L 39: "we predicted the degradation pathway" - repetition
Answer: Corrected
Comment no. 3
L 41: "improved understanding of the mechanism of phenol degradation” – this seems exaggerated. No mechanisms were studied (transcription, regulation, induction).
Answer: Corrected
Comment no. 4
L 63: Are you sure that phenol is still used as disinfectant? Please, provide citation.
Answer: Corrected. Phenol is no longer used as disinfectant. "disinfectant" has been deleted.
Comment no. 5
L 89: "involvement of key enzymes" – I could find a proof of the presence of only a single enzyme activity.
Answer: Corrected
Comment no. 6
L 166: please, define "subsystem". I would call it e.g. "functional gene group" of "gene category"
Answer: Because we are using Rapid Annotations using Subsystems Technology (RAST) for gene annotations, we believe it is best to keep the terminology as it is. A subsystem is a set of abstract functional role curated by experts
Comment no. 7
L 169: "subsystem of carbohydrates.. cofactors.." I would call it "carbohydrate metabolism, cofactor synthesis"
Answer: As explained above
Comment no. 8
L 170: antifreeze protein, cold shock proteins – please provide some comparisons with genes and enzymes from other psychrotrophic bacteria. This seems to be an interesting point.
Answer: Yes, this is interesting, as mentioned above; we are looking in detail for molecular evidence in cold-adapted bacteria for our next publication.
Comment no. 9
L 179: "The results obtained" Which results?
Answer: Reference has been added (30; Lee et al., 2018)
Comment no. 10
Figure 4: Could you provide some more data about the detected regulators and 2-component systems? Did you find what is a number of potential sigma factors? These facts deserve a short subsection.
Answer: We are planning to investigate this in future studies.
Comment no. 11
Figure 5: It would be better to put the gene or enzyme names above the arrows
Answer: Corrected
Comment no. 12
Figure 6, L 217: "… presence of enzyme-encoding genes"
Answer: Figure 6 has been deleted
Comment no. 13
L 262: "related to metabolism of amino acids"
Answer: Corrected
Comment no. 14
L 295-297: Please, rephrase
Answer: L 295-297 has been rephrased
Comment no. 15
Optimum cultivation conditions should be mentioned in Introduction, if there was a paper describing them
Answer: This is difficult since past reports of optimum Rhodococcus sp. cultivation conditions were related to specific objectives such as degradation of specific xenobiotics or fermentation process.
Comment no. 16
L 459: I guess that catechol was a substrate rather than an inducer
Answer: Changed inducer to substrate
Reviewer 2 Report
The manuscript by Lee et al. presented the studies concerning the mechanisms of phenol degradation by Rhodococcus strain. These results are of potential interest to the audience involved in developing technologies for bioremediation of hydrocarbon compounds. The paper is very interesting, well-structured, and written. The experiment was well-designed, however, the methodology and discussion are not clearly written and it needs some improvements. Overall this manuscript will be acceptable after a minor revision (below edits):
- Line 84 and line 451- please correct: g L−1” into „g/L” (all units in the manuscript should be kept consistent)
- A valuable addition would be to provide more details about the determination of phenol concentration in the “Validation of Experiments” section.
- The authors used bioinformatics tools to predict the enzymatic potential of the analyzed strain. It is well designed and well done, but I advise to mention in the discussion section that the presence of genes coding enzymes is not always connected to the expression of these genes, there are a lot of regulatory mechanisms that control the transcription processes.
- I recommend grammar-check and spell-check the paper and suggest to use language editing services to improve the paper.
Author Response
Minor comments:
Comment no. 1
Line 84 and line 451- please correct: g L−1” into „g/L” (all units in the manuscript should be kept consistent)
Answer: Corrected
Comment no. 2
A valuable addition would be to provide more details about the determination of phenol concentration in the “Validation of Experiments” section.
Answer: Reference no. 74 has been cited in the text.
Comment no. 3
The authors used bioinformatics tools to predict the enzymatic potential of the analyzed strain. It is well designed and well done, but I advise to mention in the discussion section that the presence of genes coding enzymes is not always connected to the expression of these genes, there are a lot of regulatory mechanisms that control the transcription processes. Answer: Correction done. A statement is added to note this point Comment no. 4 I recommend grammar-check and spell-check the paper and suggest to use language editing services to improve the paper. Answer: This manuscript has be proofread by a native English speaker in our research teamRound 2
Reviewer 1 Report
The comments were mostly solved by deleting the problematic parts or by a promise that the questions will by studied in future. I do not want to force the authors to make new experiments, but the uncertain points still remain. The authors hesitate to state that the detected genes are really involved in phenol degradation. In the text, the genes are called pheA, pheR, catA and so on, but the authors have no courage to put these names above the genes in Fig. 5. Instead, ORF names are given to the genes. However, it is cited, that "PheA2 and PheA1 obtained here were respectively 100% and 99.8% similar to PheA1 and PheA2" from some other Rhodococcus strain.
I consider strange that the authors measured the enzyme activity in ice. They explain that the enzyme was isolated in ice and therefore activity was measured at the same conditions to avoid losing activity. However, they did not measure the activity in any other temperature. Then, all researches should do this with every enzyme. In cultivations, maximum phenol degradation was achieved at 25 °C.
Author Response
Comment 1
The authors hesitate to state that the detected genes are really involved in phenol degradation. In the text, the genes are called pheA, pheR, catA and so on, but the authors have no courage to put these names above the genes in Fig. 5. Instead, ORF names are given to the genes. However, it is cited, that "PheA2 and PheA1 obtained here were respectively 100% and 99.8% similar to PheA1 and PheA2" from some other Rhodococcus strain.
Answer: Numbered ORFs were changed to the predicted gene designations in Fig. 5 as commented by reviewer.
Comment 2 I consider strange that the authors measured the enzyme activity in ice. They explain that the enzyme was isolated in ice and therefore activity was measured at the same conditions to avoid losing activity. However, they did not measure the activity in any other temperature. Then, all researches should do this with every enzyme. In cultivations, maximum phenol degradation was achieved at 25 °C.Answer: The screening process of phenol degradation was carried out using Plackett-Burman (PB) experiment design (Table 1 : 5-25°C).The purpose of screening process was to determine which parameters had significant effects on phenol degradation activity. Based on the screening process three parameters were selected (temperature, pH and concentration of ammonium sulphate). All three parameters were subsequently use for optimisation study using CCD. Based on the optimisation process using CCD the optimum temperature was 14.8°C which was used with the other two optimised parameters to give the best conditions for phenol degradation activity (please see Figures 1 and 2 and section 2.3).